# A LARGE-SCALE DATASET FOR ROBUST COMPLEX ANIME SCENE TEXT DETECTION

## ABSTRACT

Current text detection datasets primarily target natural or document scenes, where text typically appear in regular font and shapes, monotonous colors, and orderly layouts. The text usually arranged along straight or curved lines. However, these characteristics differ significantly from anime scenes, where text is often diverse in style, irregularly arranged, and easily confused with complex visual elements such as symbols and decorative patterns. Text in anime scene also includes a large number of handwritten and stylized fonts. Motivated by this gap, we introduce *AnimeText*, a large-scale dataset containing 735K images and 4.2M annotated text blocks. It features hierarchical annotations and hard negative samples tailored for anime scenarios. To evaluate the robustness of *AnimeText* in complex anime scenes, we conducted cross-dataset benchmarking using state-of-the-art text detection methods. Experimental results demonstrate that models trained on *AnimeText* outperform those trained on existing datasets in anime scene text detection tasks. AnimeText on HuggingFace

## 1 INTRODUCTION

In computer vision tasks, detecting and understanding textual content within images has emerged as a prominent research focus Huang et al. (2024); Duan et al. (2024); Su et al. (2024). Text detection, which focuses on localizing text regions in images, plays a crucial role in a variety of applications, including optical character recognition (OCR) Singh et al. (2021); Mishra et al. (2012), visual question answering (VQA), and vision-language chat systems Bai et al. (2025); Team et al. (2025). It is also in high demand across diverse domains such as multimedia retrieval, industrial automation, and assistive technologies for visually impaired individuals. Within the framework of multimodal large language models (MLLMs), textual understanding is an important capability, and effective text detection enhances MLLMs' ability to recognize and interpret textual content in images, along with its contextual relationships.

Existing text detection datasets Karatzas et al. (2015b); Ch'ng & Chan (2017); Singh et al. (2021) primarily focus on natural or document-centric scenes. In such scenarios, the text generally exhibits regular shapes and is typically arranged along straight or curved lines. The color of the text is usually monotonous, and its features are often clearly distinguishable from the background and other objects. Although document images contain dense textual blocks, the layout is usually well-structured, and the fonts employed are usually standardized and easily recognizable. However, these scenes differ significantly from the anime scene, which is rooted in artistic expression. This gap makes the current text detection models hard to transfer and applied to anime text detection scenarios.

Furthermore, the relatively limited scale of existing non-synthetic text detection datasets poses a challenge for the advancement of large-scale pre-trained text detection models. While larger synthetic datasets exist, there is a well-known domain gap between synthetic and real-world data, making real anime scene datasets particularly valuable for capturing the complex visual and stylistic characteristics of this domain.

Compared to natural or document scenes, anime images present a considerably more intricate and less structured scene. Text in anime often adopts highly variable forms, including numerous handwritten and stylized artistic fonts. Additionally, rather than being aligned along lines or curves, the characters are frequently scattered irregularly across the image. In contrast to natural scenes, where textual features are usually well differentiated from natural objects in terms of shape and structure, anime

images often contain a wide range of decorative elements, such as symbols, emoticons, and patterns, that may resemble text in both the color palette and artistic style. These visual similarities make detecting text more challenging. These gaps hinder the performance of existing text detection models, which are predominantly designed for natural scene text detection, when applied to anime scene.

In this paper, we propose *AnimeText*, a novel dataset for anime scene text detection, with the primary objective of enhancing text detection performance in anime scenarios. AnimeText is tailored to anime scenes and provides:

- A large-scale, high-diversity collection of high-resolution anime images containing text.

- Large-scale, high-quality text area annotations with multi-level granularity.

- Area annotations of hard negative samples that are easily confused with text.

To ensure precise and high-quality text area annotations, we design a three-stage annotation pipeline that greatly reduces workload. We further adopt a multi-round process combining pre-trained models with manual reviews to accelerate labeling. Existing datasets usually contain few images and limited text instances per image, which restricts model generalization on downstream tasks. In contrast, AnimeText provides 735k images and 4.2M annotated multilingual text instances (English, Chinese, Japanese, Korean, Russian), covering diverse text densities. On average, each image has 5.77 text instances, with some containing over 50. These characteristics make AnimeText a reliable dataset and benchmark for text detection in anime scenes.

In summary, our main contributions include:

1. A large and diverse multilanguage anime scene text detection dataset with 4.2M text annotations (5x larger than existing text detection datasets), focusing specifically on text localization tasks.

2. Extensive experiments to evaluate AnimeText showing that it is effective both as (1) a training dataset to improve anime text detection performance on multiple baseline models and (2) a testing dataset to offer a new challenge to the community.

3. A scalable annotation workflow and comprehensive analysis that provides a solid foundation for future anime text detection research.

## 2 RELATED WORK

### 2.1 TEXT DETECTION DATASETS

The field of text detection has advanced significantly through diverse datasets, from early multi-oriented English/Chinese benchmarks like MSRA-TD500 Yao et al. (2012) to large-scale general-purpose ones like COCO-Text Veit et al. (2016). Specialized datasets have emerged to address specific needs: ICDAR MLT 2017 and MLT 2019 Sanchez et al. (2017); Nayef et al. (2019) enhanced multilingual capabilities (up to ten languages including CJK); RCTW-17 Shi et al. (2017) and the extensive LSVT Sun et al. (2019) catered to Chinese text, with LSVT notable for its scale and weak annotations. Arbitrarily-shaped text detection was propelled by SCUT-CTW1500 Liu et al. (2019), Total-Text Ch'ng & Chan (2017), and ICDAR ArT Chng et al. (2019), offering polygon annotations for complex, curved text. More recently, HierText Long et al. (2022) introduced hierarchical annotations (words, lines, paragraphs) for deeper structural understanding, while video-based datasets like LSVTD Cheng et al. (2019) enabled research in dynamic text detection and tracking. Manga109 Fujimoto et al. (2016); Aizawa et al. (2020), with its Japanese manga page annotations (mainly speech bubbles), is relevant but distinct from anime's needs.

Existing text detection datasets, despite advancements, lack a benchmark for anime. Manga109 focuses on scanned comic pages, which presents static layouts, while anime scenes involve dynamic compositions, motion blur, and stylized elements. Other datasets designed for natural images and don't generalize to anime's unique visuals, leading to poor performance on anime images. Our AnimeText dataset fills this gap with a large, high-quality benchmark, offering detailed annotations across diverse anime styles and text types (subtitles, signs, effects).

## 2.2 OCR DATASETS

OCR datasets are fundamental to text recognition research and vary in terms of text layout and annotation granularity. Benchmarks such as IIIT5K-Words Mishra et al. (2012) and SVT Wang et al. (2011) focus on regular, horizontal text, whereas IC15 Karatzas et al. (2015a) and CUTE80 Risnumawan et al. (2014) address irregular and curved text. Many text detection datasets, including the multilingual MLT 2019 Nayef et al. (2019), large-scale Chinese-focused LSVT Sun et al. (2019), and arbitrarily-shaped ArT Chng et al. (2019), also serve as comprehensive end-to-end OCR benchmarks. More recent initiatives such as TextOCR Singh et al. (2021) provide large-scale annotations for diverse real-world scenes, while domain-specific datasets like ReCTS Zhang et al. (2019) offer character-level annotations for Chinese signboard text. Synthetic datasets have also been instrumental in pretraining OCR models, evolving from SynthText Gupta et al. (2016) and Synth90k Jaderberg et al. (2014) to multilingual SynthTIGER Yim et al. (2021), which supports Japanese text rendering. For Japanese text in comics, Manga109 Fujimoto et al. (2016); Aizawa et al. (2020) provides manually transcribed content from static manga pages.

Despite the diversity and scale of these datasets, recognizing text in anime introduces unique challenges that are not fully addressed. AnimeText provide a large scale of text area annotations for anime images, which can support for the community to build anime OCR datasets efficiently.

## 2.3 DOWNSTREAM APPLICATION MODELS

In recent years, the field of scene text detection and recognition has witnessed significant advancements, with a series of innovative methods and models emerging that offer new insights and tools for text processing. ODMDuan et al. (2024) proposed a destylization-based pre-training method for OCR, unifying diverse text styles to improve text–image alignment, especially for varied fonts. Bridging Text Spotting Huang et al. (2024) connects fixed detectors and recognizers with a zero-initialized network, combining modularity with end-to-end optimization for efficient multilingual text spotting. LRANet Su et al. (2024) applies low-rank approximation and regression to fit polygonal text boundaries, reducing complexity and inference time without sacrificing accuracy.

However, despite these advancements, their performance often declines when applied directly to the unique conditions of anime-style images. This discrepancy largely stems from the relative scarcity of anime data in mainstream training corpora. General scene-text datasets differ markedly from anime imagery in CJK character morphology, background complexity, colour palettes, and artistic stylisation. Consequently, models trained primarily on such data may not fully generalise to anime-specific visual characteristics and text presentation styles. This underscores the importance of developing and leveraging datasets containing high-quality anime text images, such as the *AnimeText* dataset proposed herein, to improve downstream models in this specific and challenging domain.

## 3 THE CHALLENGES OF TEXT DETECTION ON ANIME SCENE

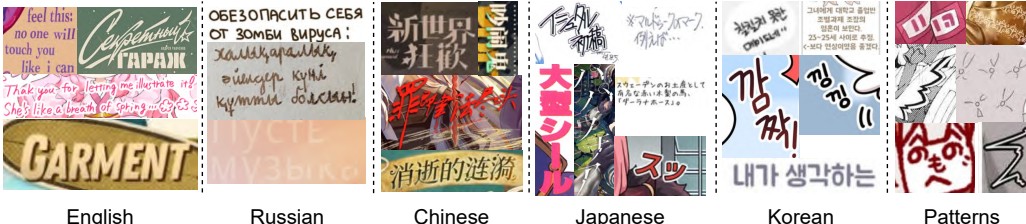

| English | Russian | Chinese | Japanese | Korean | Patterns |

Figure 1: Multilingual text and confusing patterns examples in anime scene.

Existing natural scene text detection datasets Singh et al. (2021); Karatzas et al. (2015b); Sun et al. (2019); Ch'ng & Chan (2017) usually contain only one or two languages (e.g., English or Chinese) and have limited linguistic and stylistic diversity. Whether printed or handwritten, text in natural scenes is primarily designed for readability rather than decoration, favoring regular fonts and consistent layouts. In real-world scenes, most human-made objects and natural landscapes differ greatly from text, and text-like patterns are rare. Although lens distortion or perspective may affect appearance,

the relative spatial arrangement of characters usually remains stable, with limited impact on intrinsic features.

In contrast, anime scenes often contain multilingual text with diverse forms and characteristics. Not all text is mainly intended for readability; many instances are artistic, serving decorative purposes and featuring diverse, often irregular styles and layouts. Additionally, anime images contain numerous symbols and patterns that visually resemble text and can be easily mistaken for characters. As illustrated in Figure 1, viewers unfamiliar with the language may find it difficult to distinguish these patterns from actual text. Anime scenes contain numerous decorative elements, with text as just one component, complicating text and non-text differentiation compared to natural scenes. Models trained on natural scene text detection datasets often fail to detect such complex text and are prone to misclassify decorative patterns as text. The limited performance of existing models in anime text detection scenarios highlights the need for a dataset specifically designed for anime scenes with its unique characteristics.

# 4 ANIMETEXT DATASET BUILDING

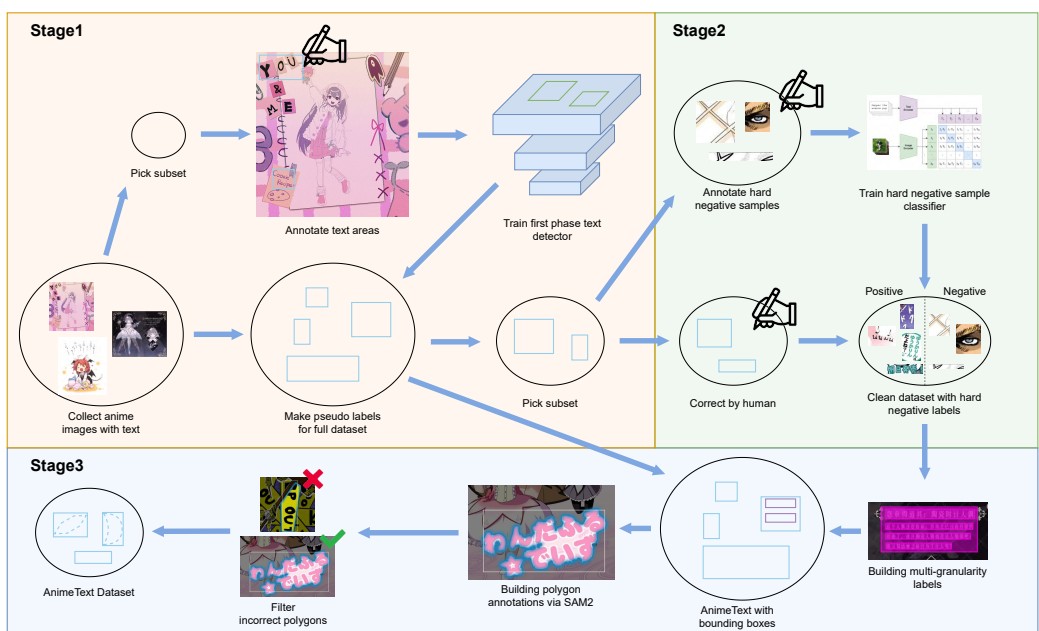

Figure 2: Pipeline to building the AnimeText.

In this section, we present the details on the collection, annotation, and quality of the dataset respectively.To ensure dataset diversity, we gathered images from multiple online sources, all of which are compliant with research-use licenses. To construct a reliable dataset while minimizing manual annotation costs, we designed a three-stage annotation pipeline.

## 4.1 STAGE 1: MANUAL ANNOTATION

In the first stage, we manually selected images containing text from a large collection of anime images. From these, we make a subset of approximately 50,000 images for box-level annotation. The annotations were carried out by a team, with the majority of annotators responsible for the initial bounding box annotation and few annotators responsible for reviewing and correcting the bounding boxes. Annotations of text blocks were performed at the level of semantically coherent text blocks (e.g., a closely arranged sentence), rather than at the word level as in existing datasets Karatzas et al. (2015b); Singh et al. (2021). The definition of a text block $\mathbf{B}_l$ is as follows:

$$\forall c_i \in \mathbf{B}_l, \exists c_j \in \mathbf{B}_l, \quad d(c_i, c_j) \leq \alpha \cdot \min(\bar{w}(B_l), \bar{h}(B_l))$$
$$\forall c_k \notin \mathbf{B}_l, \forall c_i \in \mathbf{B}_l, \quad d(c_i, c_k) > \alpha \cdot \min(\bar{w}(B_l), \bar{h}(B_l)) \tag{1}$$

where $\bar{w}(B_l) = \frac{1}{n}\sum_{i=1}^{n} c_i^w$ and $\bar{h}(B_l) = \frac{1}{n}\sum_{i=1}^{n} c_i^h$ denote the average width and height of all words in $\mathbf{B}_l$, $d(\cdot, \cdot)$ denotes the distance between two words, and $\alpha$ is the threshold for text blocks. In our annotation, $d(\cdot, \cdot)$ is defined as the distance between the closest edges of two character bounding boxes, and $\alpha$ is set to approximately 0.5, acknowledging potential variance due to manual labeling. Bounding boxes are required to tightly enclose the text regions, ensuring no characters are omitted and no superfluous areas are included.

We accelerated the initial, single-granularity bounding box annotation via a semi-automated pipeline. A preliminary YOLOv11 model, trained on a multilingual subset (50 epochs, Adam optimizer, lr=1e-3), generated pseudo-labels for the entire dataset with a 0.45 confidence threshold, followed by manual verification and correction.

## 4.2 STAGE 2: ANNOTATE HARD NEGATIVE SAMPLES

**Failure Cases Analysis**

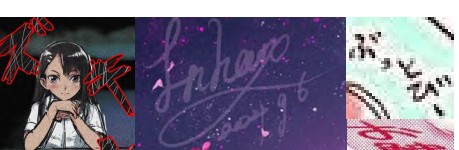 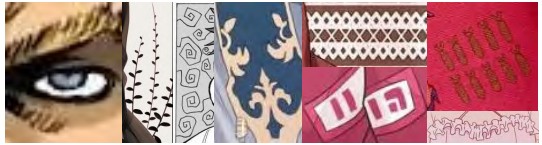

Missing Samples            Hard Negative Samples

Figure 3: Examples of missing samples and hard negative samples. Missing samples: Text instances that were not annotated during Stage 1 (false negatives); Hard negative samples: Background elements that were mistakenly identified as text during Stage 1 (false positives).

In the first stage of annotation, we analyzed the pseudo-labels that were incorrectly predicted by the YOLO model Khanam & Hussain (2024), as illustrated in Figure 3. These incorrect pseudo-labels can be classified into the following categories: (1) Exaggerated or stylized artistic text; (2) Text with low contrast against the background; (3) Patterns or symbols resembling textual features; (4) Text regions with disordered character arrangements. The first two categories are often omitted by the model as they tend to be misclassified as background elements, whereas the third category is frequently misidentified as text. The fourth category typically results in inaccurate bounding boxes that fail to correctly enclose the text regions. We designate the bounding boxes of errors in categories 3 and 4 as Hard Negative Samples. These samples pose significant challenges for anime text detectors, and if not explicitly labeled during training, can negatively impact model performance. Anime scenes often contain a large number of artistic text elements and text-like patterns or symbols. A robust anime text detection model must be capable of accurately distinguishing between these two types of elements.

**Hard Negative Sample Classifier**

To efficiently annotate Hard Negative Samples, we selected a subset from the AnimeText dataset and manually labeled all Hard Negative Samples, as well as text samples belonging to categories 1 and 2, which are easily confused. These Hard Negative Samples are then used to fine-tune a pretrained CLIP Radford et al. (2021) image encoder, enhancing its ability to discriminate between actual text and Hard Negative Samples. Our Hard Negative Sample classifier is fine-tuned based on the CLIP-H model. To better leverage the pretrained knowledge and retain its feature extraction capabilities, we fine-tune only the 26th to 32nd layers of the image encoder $\mathcal{E}_\theta$, and append a classifier head $f_\phi$ at the end. The classifier head weights $\phi$ are initialized using the embeddings generated by the CLIP text encoder with [text, pattern] input, and head forward is performed using cosine similarity:

$$p = \frac{\phi \cdot \mathcal{E}_\theta(x)}{\|\phi\| \cdot \|\mathcal{E}_\theta(x)\|} \tag{2}$$

where $x$ is the input image and $p$ denotes the probability that $x$ is a text instance. The Hard Negative Sample classifier enables the annotation of Hard Negative Samples across the entire dataset, thereby facilitating more effective training and evaluation of anime text detection models.

**Classifier Performance Evaluation**

The CLIP-H-based classifier was fine-tuned on 491 text patches and 473 hard negative patches, and evaluated on 150 held-out samples. The classification performance is shown in Table 1.

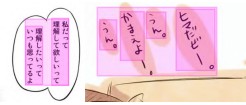 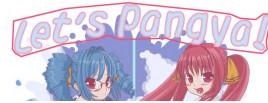 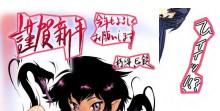

Figure 4: Hierarchical annotations examples in AnimeText.

Figure 5: Polygon annotations examples in AnimeText.

Table 2: Image and text instants count of our AnimeText and other natural scene text detection dataset.

| Dataset | Images | | | Instances | | |
|---|---|---|---|---|---|---|
| | Train | Vaild | Test | Train | Vaild | Test |
| CTW-1500 | 1000 | - | 500 | 2774 | - | 3575 |
| Total-Text | 1255 | - | 300 | 9276 | - | 2215 |
| COCO-Text | 18895 | - | 4416 | 61793 | - | 13910 |
| ICDAR19-ArT | 5603 | - | 4563 | 50042 | - | - |
| TextOCR | 24902 | - | 3232 | 822572 | - | 80497 |
| AnimeText (Ours) | 514144 | 147191 | 73725 | 3228144 | 922758 | 88320 |

Applying this classifier to filter pseudo-labels significantly reduced false positives, leading to a 26.9% increase in precision. As illustrated in Figure 3, most first-stage errors were background regions misclassified as text, which the classifier effectively suppressed.

Unlike the background elements in natural scenes, which in anime scenes have a substantial number of Hard Negative Samples that closely resemble text. Accurate annotations of these samples are essential for designing and training reliable anime text detection models.

Table 1: Hard Negative Sample Classifier performance.

| Accuracy (%) | F1-score (%) |
|---|---|
| 98.4 | 98.1 |

### 4.3 STAGE 3: BUILDING MULTI-GRANULARITY ANNOTATIONS

**Hierarchical Annotations**

In certain anime scenes, text may appear in long passages or be scattered irregularly. To address varying requirements, we construct hierarchical multi-granularity annotations. According to the definition of text blocks in Section 4.1, a text block typically consists of a row or column of text, or a tightly arranged group of characters. This constitutes the finest level of annotation, denoted as $B^0$. Based on the annotations in $B^0$, coarser-grained text annotations can be constructed. We consider that a set of closely arranged $B^0$ boxes can form a coarser-grained annotation $B^1$, which is formally defined as:

$$\forall B_i^0 \in \mathbf{B}_l^1, \exists B_j^0 \in \mathbf{B}_l^1, \quad d(B_i^0, B_j^0) \leq \beta \cdot \min(\bar{w}(B_l^1), \bar{h}(B_l^1))$$
$$\forall B_k^0 \notin \mathbf{B}_l^1, \forall B_i^0 \in \mathbf{B}_l^1, \quad d(B_i^0, B_k^0) > \beta \cdot \min(\bar{w}(B_l^1), \bar{h}(B_l^1))$$

(3)

Figure 4 illustrates the hierarchical, multi-granularity text annotations, where smaller text boxes correspond to $B^0$, and those enclosing multiple $B^0$ boxes correspond to $B^1$. In MLLMs, multi-granularity text detection enables more accurate localization of relevant text regions, thereby enhancing the model's ability to comprehend textual content within images.

**Polygon Annotations with Segmentation Model**

Box-level annotations are often imprecise for distorted text, capturing excessive background that impairs downstream tasks like scene text recognition. We therefore introduce finer-grained polygon annotations to provide a tighter text boundary. To efficiently construct this dataset, we employ a semi-automated pipeline: the Segment Anything Model (SAM) generates initial polygon proposals from existing box annotations, which are then manually refined for high quality, as illustrated in Figure 5. This enables efficient construction of the polygon-level annotated AnimeText dataset.

## 5 STATISTICS AND ANALYSIS OF ANIMETEXT DATASET

**Strength in Scale.** Table 2 compares the number of images and annotated text instances in AnimeText with existing text detection datasets. Whereas TextOCR and Total-Text annotate individual words

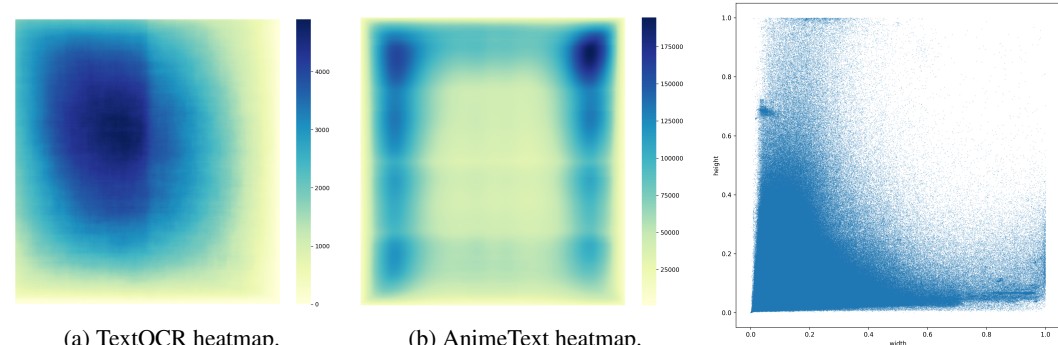

(a) TextOCR heatmap.   (b) AnimeText heatmap.

Figure 6: Word locations' heatmaps across images with blue indicating higher density.

Figure 7: Visualization of the size distribution of text instances in AnimeText.

as text instances (mainly for OCR), AnimeText adopts a coarser annotation strategy by labeling text blocks as instances. With 735k images, AnimeText is 26.12× larger than TextOCR and 72.3× larger than ICDAR19-ArT. In terms of the number of text instances, AnimeText also exhibits a significantly larger scale, containing 4.2M instances, which is 4.69× that of TextOCR and 84.71× that of ICDAR19-ArT, with an average of 5.77 text instances per image. This large-scale images and text annotations offers a solid foundation for training and evaluating text detection models in anime scenes, even exceeding the scale of natural scene datasets.

**Text Spatial Distribution.** Figure 6 illustrates the spatial distribution of text bounding boxes in AnimeText relative to the image, where darker blue regions indicate a higher density of text bounding boxes. Empirical observations indicate that the spatial distribution of text in anime scenes exhibits significant differences from that in natural scenes. In anime images, most text instances are located along the periphery, whereas in natural images, they tend to be concentrated near the center. This is likely due to the most artist prefer to put the characters and other elements in the center. Such a distributional disparity may partially account for the poor performance of existing natural scene text detection models on anime scenes.

Figure 7 presents the distribution of text instance sizes as a percentage of the image. It can be seen that the bounding boxes in AnimeText span a wide range of sizes percentage, with images containing text instances at diverse scales. This offers a robust data foundation for enhancing the generalizability of text detection models. Specifically, it can facilitating model learning to detect text at varying scales and granularities, while also facilitating the construction of reliable benchmarks for evaluating model performance on multi-scale and proportionally varied text detection tasks.

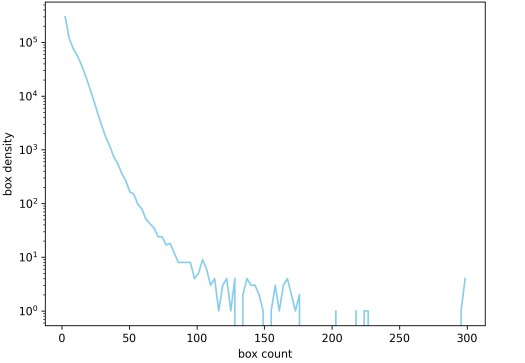 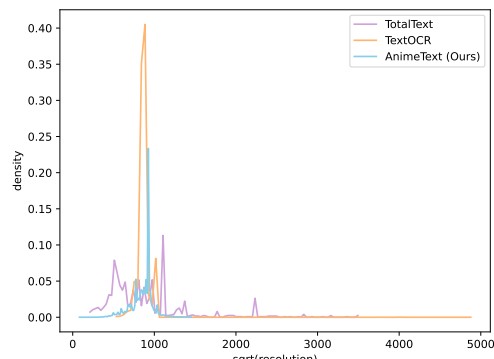

Figure 8: Text instances density of AnimeText.

Figure 9: Image resolution density of AnimeText and other natural text detection datasets.

**Text Density.** Images with high text density facilitate text detection models to effectively learn to distinguish text from other visual elements. Figure 8 presents the distribution of text density in AnimeText, compared to that in existing natural scene text detection datasets. The results show that

Table 3: Mean and standard deviation statistics for the natural scene dataset and the anime dataset.

| Dataset | Mean | | | Std | | |
|---|---|---|---|---|---|---|
| | Red | Green | Blue | Red | Green | Blue |
| ImageNet | 0.485 | 0.456 | 0.406 | 0.229 | 0.224 | 0.225 |
| Total-Text | 0.457 | 0.427 | 0.402 | 0.284 | 0.277 | 0.284 |
| AnimeText (Ours) | 0.677 | 0.618 | 0.612 | 0.317 | 0.319 | 0.315 |

while most AnimeText images contain fewer than 10 text instances, a substantial number of images feature between 50 and 100 instances, with some containing more than 100, shown in Figure 10.

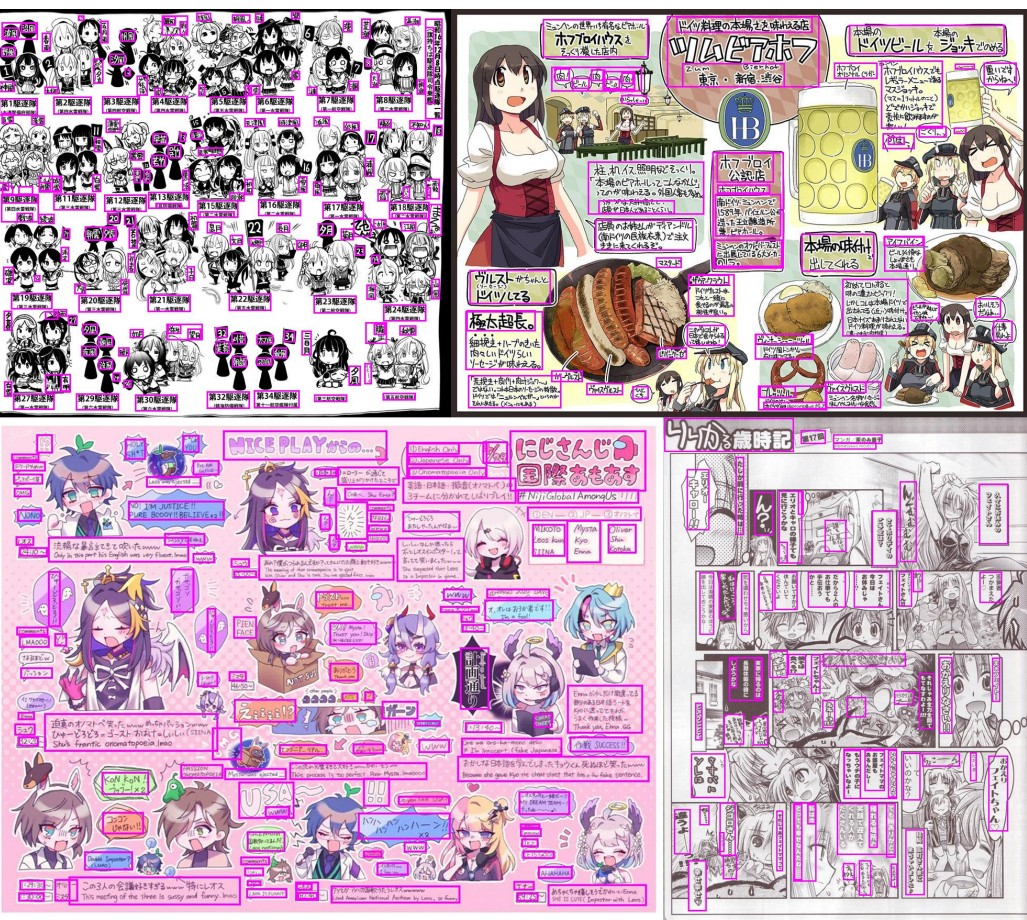

Figure 10: Examples of images with high text density in anime scene.

This high stylistic variability and density of text in anime scenes distinguishes our dataset from natural-scene counterparts. Images with high text density (e.g., 50+ instances) are common, featuring complex spatial distributions and a combination of multilingual text, handwritten fonts, and stylized lettering often interwoven with decorative patterns. This inclusion of complex, high-density scenarios is crucial for training models that can generalize to intricate layouts where text and visual elements are densely interwoven.

**Resolution.** The AnimeText dataset provides high-resolution images comparable to those existing natural scene datasets, as illustrated in Figure 9. Text detection on high-resolution images enables more precise localization of textual elements and can work well in the real-world deployment of multimodal large models. Most images in AnimeText have resolutions around $900^2$, while also covering a diverse range of resolutions suited for typical high-resolution use cases.

**Image Distribution Analysis.** The color distribution of anime scene images differs substantially from that of natural scenes. As shown in Table 3, the mean and standard deviation observed in the ImageNet dataset closely match those of TotalText, suggesting a similar underlying distribution for natural scene datasets. In contrast, the anime scene images in AnimeText exhibit a significantly

different statistical distribution. Compared to natural images, anime images have higher means and larger variances, suggesting that anime images typically have higher saturation and contrast.

**Multilingual Coverage Analysis.** AnimeText provides comprehensive multilingual support with text instances spanning five major languages. The language distribution is shown in Table 4, with Japanese being the dominant language (65.57%), followed by English (30.21%). This distribution reflects the anime industry's linguistic landscape and provides a realistic foundation for multilingual text detection research.

Table 4: Language distribution in AnimeText dataset

| Language | English | Russian | Chinese | Japanese | Korean | Others |
|---|---|---|---|---|---|---|
| Percentage (%) | 30.21 | 0.30 | 1.44 | 65.57 | 0.62 | 1.86 |

The diverse linguistic representation enables comprehensive evaluation of text detection models across different writing systems, including Latin (English, Russian), logographic systems (Chinese, Japanese), and syllabic (Korean). While artistic fonts in anime scenes are difficult to quantify systematically, AnimeText includes stylistic diversity anime content across these languages.

## 6 EXPERIMENTS

### 6.1 CROSS-DATASET EVALUATION

Observe from experiments in Table 5, a significant domain gap exists for text detection between natural and anime scenes. Models pre-trained on natural scene datasets like ICDAR15 Karatzas et al. (2015b), such as Bridging Text Spotting Huang et al. (2024), DBNet, and LRANet, exhibit a severe performance degradation on anime text, with F1-scores dropping to as low as 0.008. To address this, we introduce the `AnimeText` dataset. Training on `AnimeText` effectively bridges this gap, enabling text-specific detectors like DBNet and LRANet to achieve competitive F1-scores of 0.743 and 0.855, respectively. Conversely, models trained exclusively on anime data also perform poorly on natural scenes, confirming a substantial discrepancy in textual characteristics. These benchmarks for anime text detection task, validated across diverse architectures, demonstrate the effectiveness and utility of our proposed dataset.

### 6.2 ABLATION STUDY ON HARD NEGATIVE SAMPLE FILTERING

To evaluate the effectiveness of our hard negative sample filtering approach, we conducted ablation studies using YOLO v11 as the baseline model. As shown in Table 6, the inclusion of hard negative sample filtering significantly improves model performance.

As illustrated in Figure 3, most incorrect annotations in the first-stage pseudo-labeling are hard negative samples—background regions misclassified as text. Annotating and filtering these significantly

Table 5: Cross dataset evaluation on natural and anime scene. The results indicate that the domain gap between anime scenes and natural scenes is significantly larger than that between natural scenes.

| Method | Train Dataset | Test Dataset | Precision | Recall | F1-score | mAP$_{50:95}$ |
|---|---|---|---|---|---|---|
| YOLO v11 | ICDAR15 | ICDAR15 | 0.657 | 0.495 | 0.565 | 0.291 |
| YOLO v11 | AnimeText | ICDAR15 | 0.187 | 0.31 | 0.233 | 0.083 |
| YOLO v11 | ICDAR15 | AnimeText | 0.0629 | 0.106 | 0.079 | 0.01 |
| YOLO v11 | AnimeText | AnimeText | 0.878 | 0.825 | 0.851 | 0.806 |
| YOLO v11 | CTW1500 | CTW1500 | 0.8072 | 0.7901 | 0.7986 | 0.5503 |
| YOLO v11 | CTW1500 | AnimeText | 0.2335 | 0.2673 | 0.2493 | 0.0698 |
| YOLO v11 | AnimeText | CTW1500 | 0.4224 | 0.3238 | 0.3666 | 0.1754 |
| YOLO v11 | CTW1500 | ICDAR15 | 0.4579 | 0.3722 | 0.4106 | 0.1661 |
| YOLO v11 | ICDAR15 | CTW1500 | | | | |
| DBNet | ICDAR15 | AnimeText | 0.022 | 0.005 | 0.008 | - |
| DBNet | AnimeText | AnimeText | 0.851 | 0.660 | 0.743 | - |
| LRANet | ICDAR15 | AnimeText | 0.124 | 0.145 | 0.134 | - |
| LRANet | AnimeText | AnimeText | 0.833 | 0.878 | 0.855 | - |
| Bridging Text Spotting | ICDAR15 | AnimeText | 0.035 | 0.189 | 0.056 | - |

Table 6: Ablation study on hard negative sample filtering

| Method | Precision | Recall | mAP |
|---|---|---|---|
| w/o hard negative sample filtering | 0.668 | 0.766 | 0.701 |
| hard negative sample filtering | 0.857 | 0.819 | 0.756 |
| hard negative sample reweighting | 0.878 | 0.825 | 0.806 |

reduced false positives and improved precision by 26.9%. Following prior work, we applied hard negative sample reweighting during training, which further improved performance. This analysis highlights the importance of our data-centric design for anime text detection.

### 6.3 IMPROVING THE STATE-OF-THE ART

To evaluate the effectiveness of our proposed AnimeText dataset, we evaluate based on the YOLO v11 model. The model is trained for 50 epochs using a batch size of 512, the AdamW optimizer, with a learning rate of 0.001, on 1 A100 GPU in 26h. As shown in Table 5, the YOLO v11 model trained on AnimeText significantly outperforms existing state-of-the-art methods in anime scene text detection, achieving more accurate localization of text regions. The $mAP_{50:95}$ results further validate that the model trained on AnimeText can precisely identify textual areas within anime images.

## 7 CONCLUSION

In this work, we introduce *AnimeText*, a large-scale dataset for anime scene text detection. It provides multilingual annotations, hierarchical bounding boxes, and hard negative samples to advance precise text localization, deliberately decoupling detection from recognition. The hierarchical annotations are designed to support diverse downstream applications, such as manga restoration and text translation.

To facilitate future research, we provide comprehensive baselines and benchmarks. We anticipate that *AnimeText* will help bridge the gap between natural and anime scene text detection and inspire further advancements, particularly for downstream Multimodal LLM (MLLM) applications.

**Limitations and Future Work.** Our dataset currently focuses on static anime scenes as a foundational step toward reliable anime text detection, leaving the extension to dynamic scenes with temporal annotations as a future direction. AnimeText focuses on text detection, while ensuring localization accuracy, limits its direct use for end-to-end OCR and VQA tasks. We plan to address this by adding text transcriptions in future work to create a comprehensive anime OCR benchmark. Future research may also benefit from investigating universal text detection datasets that are generalizable across both anime and natural scenes.

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
