# A Large-scale Dataset for Robust Complex Anime Scene Text Detection (Supplementary Material)

## 1 Density Stratified Analysis

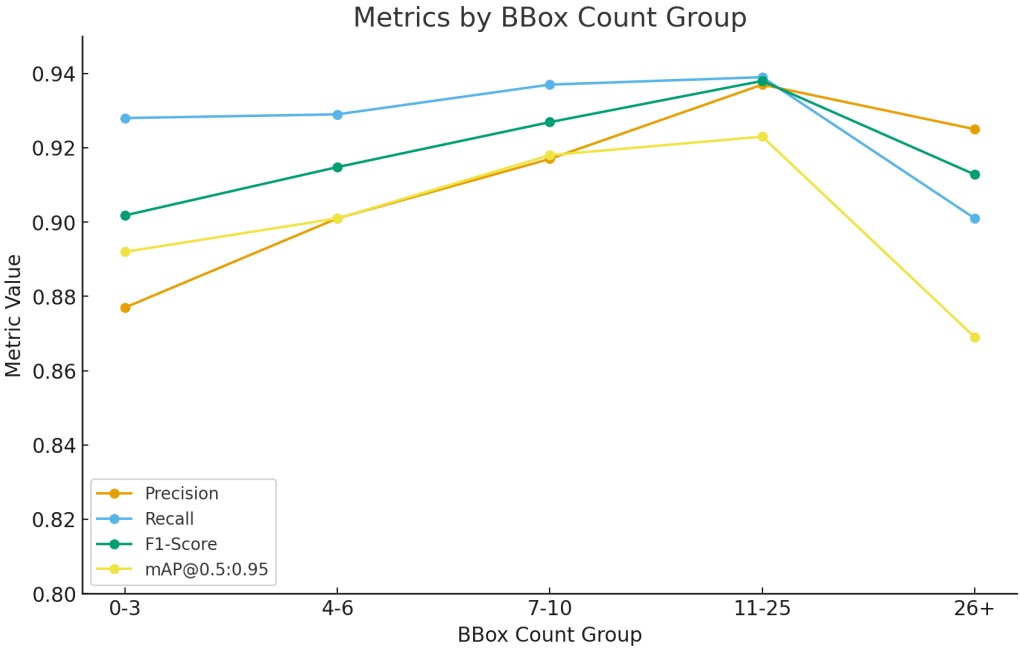

Figure I: Density-stratified dataset evaluation.

Figure I shows the density-stratified analysis, measuring anime text detector performance across images with varying text densities. The resulting curves show that while performance slightly decreases for extremely high-density images, the detection performance remains relatively stable across most density levels. This demonstrates the robustness of the dataset and baseline models against varying text distributions.

## 2 Data Diversity Analysis

AnimeText dataset explicitly includes a wide variety of real-world and user-generated content (as shown in Figure II), including screenshots, edited anime images, comics, posters, derivative fan works, and AI-generated images. This diversity is a key advantage of AnimeText over existing manga or comic datasets, which often rely on clean, scanned pages.

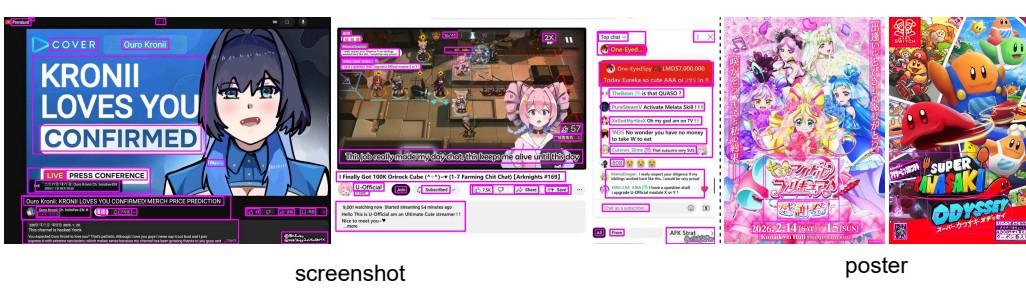

screenshot

poster

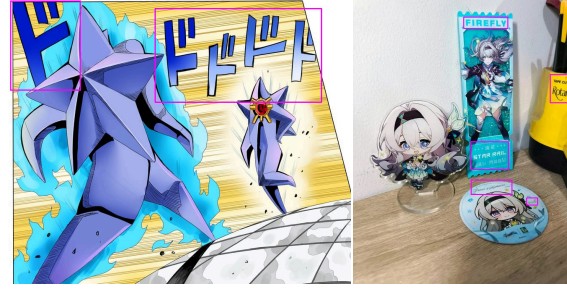

edited anime images /
real-world mixed images

derivative / fan work

Figure II: Examples of images with high text density in anime scene.