# OpenReview forum: "A Large-scale Dataset for Robust Complex Anime Scene Text Detection"
_ICLR.cc/2026/Conference — Submitted to ICLR 2026_

### Official Review · Reviewer_7oZU · 2025-10-27

**Soundness:** 3
**Presentation:** 2
**Contribution:** 2
**Rating:** 4
**Confidence:** 3

**Summary:**

This paper introduces AnimeText, a large-scale dataset designed for text detection in anime scenes, aiming to fill the gap between existing scene text detection datasets (mostly focusing on natural or document images) and the unique visual characteristics of anime imagery.
The dataset contains 735K images and 4.2M annotated text blocks, covering five major languages (English, Chinese, Japanese, Korean, Russian). The authors propose a three-stage annotation pipeline combining semi-automatic detection (YOLOv11 pseudo-labels), a CLIP-based hard negative sample classifier, and SAM-assisted polygon refinement. Comprehensive statistical analyses and cross-dataset experiments show that models trained on AnimeText outperform those trained on existing natural scene datasets in anime text detection tasks. Overall, the paper aims to contribute a benchmark dataset rather than a novel model or algorithm.

**Strengths:**

1. **Clear motivation and relevance.**
The paper identifies a realistic and under-explored problem domain — text detection in anime scenes — which differs visually and stylistically from natural scenes. The motivation is well aligned with the Datasets & Benchmarks track.

2. **Scale and diversity.**
AnimeText is significantly larger than existing datasets (4.2M instances, 735K images) and includes multilingual text with diverse fonts, densities, and artistic styles. This scale can support downstream text detection and OCR research.

3. **Systematic annotation pipeline.**
The three-stage annotation process (YOLO → CLIP → SAM) is well-engineered and reproducible, balancing automation and human verification. Introducing hard negative samples explicitly is a valuable design choice that may improve model robustness.

**Weaknesses:**

1. **Incomplete discussion of existing datasets.**
The claim that “no anime text detection datasets exist” is not entirely accurate.
Prior works such as Manga109, Manga-Text-Detection, and other manga-oriented datasets already include bounding-box or polygon annotations for stylized text.
The authors should explicitly differentiate “anime scene frames” from “manga/comic pages” and summarize all related datasets in a comparative table (language coverage, scene type, annotation type, scale, etc.).

2. **Weak justification of domain uniqueness.**
The cross-dataset experiment (training on natural scenes → testing on AnimeText, and vice versa) only demonstrates a domain gap, not necessarily that anime is a “unique” domain.
Any two datasets from different sources would likely show similar degradation.
To claim anime as a distinct domain, the authors should include an additional control experiment (e.g., natural dataset A → B vs. AnimeText → natural) to show the gap is significantly larger.

3. **Insufficient evidence for MLLM relevance.**
Although the introduction repeatedly mentions benefits for multimodal LLMs, all experiments are conducted with traditional detection models (YOLOv11, DBNet, LRANet).
No multimodal or downstream experiments (e.g., OCR recognition, image-text QA) are provided. The claim that AnimeText will “enhance MLLMs” is therefore overstated.
The authors should either include a small-scale demonstration or tone down the claim.

**Questions:**

1. Can the authors clarify the difference between “anime scene” and existing manga text datasets (e.g., Manga109)?
Are there examples where Manga109-trained models fail but AnimeText-trained models succeed?

2. How large is the observed cross-domain gap compared to gaps between other natural scene datasets?
Would similar degradation appear when training on TextOCR and testing on COCO-Text?

3. Could the authors show any downstream example (e.g., OCR recognition, subtitle translation, or VQA) to justify the claimed relevance for multimodal LLMs?

---

> ### Author Response · Authors · 2025-11-25
>
> ## W1+Q1
>
> **Q: Incomplete discussion of existing datasets.**
>
> A: We clarify that as the first anime text detection dataset, adapting SOTA text detectors serves as a rigorous baseline. While existing manga datasets focus on static, black-and-white content, our dataset acts as a **superset** with a significant ​**domain gap**​:
>
> 1. **Content Diversity:** Beyond comics, our dataset integrates AI-generated ​**anime images**​, screenshots, **edited anime, ​**posters, and derivative fan works.
> 2. **Visual Gap:** Unlike **monochromatic** manga scans, anime images usually feature complex ​**RGB color spaces**​, dynamic ​**lighting/shading**​, and video-specific anomalies like **motion blur** and compression artifacts.
> 3. **Text Layout:** Anime text can lacks speech bubbles and features extreme artistic stylization (e.g., heavy strokes, glow effects).
>
> These unique challenges highlight why existing manga-based methods are insufficient and underscore the necessity of our proposed benchmark.
>
> We further provide the comparison of AnimeText with manga109 (SOTA manga dataset):
>
> | Train Dataset | Test Dataset | Precision | Recall | F1-score | mAP@0.5:0.95 |
> | ------ | ------- | ------- | -------- | ---------- | -------------- |
> | Manga109      | Manga109     | 0.9453    | 0.9336 | 0.9394   | 0.8215       |
> | Manga109      | AnimeText    | 0.8646    | 0.3985 | 0.5455   | 0.3742       |
> | AnimeText     | Manga109     | 0.8441    | 0.8966 | 0.8696   | 0.6912       |
> | Manga109      | CTW1500      | 0         | 0      | 0        | 0            |
> | Manga109      | ICDAR15      | 0         | 0      | 0        | 0            |
>
> As shown in table above, a model trained on Manga109 achieves **high Precision (0.86)** but **low Recall (0.40)** on AnimeText. This leads to two conclusions:
>
> 1. **AnimeText is a Superset:** The high precision confirms our dataset covers standard clean data, while the low recall proves it extends significantly beyond it. The \~60% missed instances correspond to the complex, real-world and user-generated images (e.g., fan-edits, distortions) unique to AnimeText.
> 2. **Robustness:** The manga109 trained model fails to generalize on natural scenes (CTW1500/ICDAR15), confirming that AnimeText's diverse distribution is essential for robust, practical generalization.
>
> ## W2+Q2
>
> __Q: Weak justification of domain uniqueness.__
>
> A: Thanks for your suggestions. We have performed additional experiments using YOLO v11 to compare CTW-1500, ICDAR15, and AnimeText.
>
> Results in Table 5 in the revised main paper support two key conclusions:
>
> 1. **Consistency with Table 5:** A significant domain gap exists between natural and anime scenes. The model trained on CTW-1500 performs poorly on AnimeText (F1 0.2493).
> 2. **Domain Gap Magnitude:** While a domain gap exists between natural datasets (CTW-1500 \\(\to\\) ICDAR15, F1 0.4106), it is significantly smaller than the gap between natural and anime datasets (CTW-1500 \\(\to\\) AnimeText, F1 0.2493).
>
> ## W3+Q3
>
> **Q: The claim that AnimeText will “enhance MLLMs” is therefore overstated. The authors should either include a small-scale demonstration or tone down the claim.**
>
> A: For enhance MLLMs, we highlight two key advantages supported by our experiments:
>
> 1. **Facilitating Speaker-Text Association:** Unlike standard documents, anime requires linking speech bubbles to specific characters. Precise bounding boxes provide the necessary **spatial grounding** to map text to speakers (Speaker-Text Association). This structural information is vital for narrative understanding, an area where end-to-end MLLMs often struggle due to complex layouts.
> 2. **Mitigating Contextual Interference:** Anime images often feature cluttered backgrounds, diverse artistic styles, and complex layouts that can distract MLLMs during full-image inference. Our experiments demonstrate that a **"Detect-then-Crop"** strategy effectively removes this background noise and layout interference. This allows recognizers (e.g., DeepSeek-OCR) to focus exclusively on the text features, significantly improving accuracy compared to end-to-end processing. As shown in the table below, using our detector to crop text regions acts as a "hard attention" mechanism, drastically reducing the Normalized Edit Distance (NED) from **0.2961** to ​**0.1030**​. This proves that explicit detection is essential to filter out anime background noise and allow recognizers to focus on text features.
>
> | Method   | NED |
> | -- | -- |
> | DeepSeek-OCR  | 0.2961 |
> | DeepSeek-OCR w/ AnimeText crop | 0.1030 |

---

> > ### Comment · Reviewer_7oZU · 2025-11-27
> > **Reviewer Comment**
> >
> > Thank you for the detailed rebuttal and clarifications.
> > After carefully reviewing the additional explanations and evidence, my evaluation remains unchanged.
> > I appreciate the authors’ efforts during the rebuttal phase.

---

> ### Author Response · Authors · 2025-11-28
> **Concern Regarding Generic Feedback and Lack of Justification**
>
> Dear Reviewer 7oZU,
>
> We are disappointed by your post-rebuttal response. The statement provided is entirely **generic** and could apply to any paper in any field. It does not address a single specific point from our detailed rebuttal, nor does it justify why our clarifications were insufficient.
>
> As this is a dataset paper, we provided concrete evidence regarding the utility and annotation quality of our work. **Dismissing these efforts with a template-like response without providing technical grounds undermines the purpose of the ICLR discussion phase.**
>
> If you maintain your negative score, **we respectfully request, as per ICLR reviewer guidelines, that you provide specific, technical reasons for doing so.** Without such justification, your evaluation appears arbitrary and provides no basis for the Area Chair to make an informed decision.
>
> Best regards, The Authors.

---

### Official Review · Reviewer_wHE5 · 2025-10-31

**Soundness:** 3
**Presentation:** 2
**Contribution:** 2
**Rating:** 2
**Confidence:** 4

**Summary:**

The paper presents AnimeText, a large-scale dataset containing 735K images and 4.2M text annotations for anime scene text detection. While the work addresses an underexplored domain with substantial data collection efforts, the paper suffers from limited task scope (detection-only without OCR capability), insufficient experimental validation, and lack of depth in analyzing what makes anime text unique.

**Strengths:**

1. The dataset addresses the overlooked domain of anime scene text detection, which has distinct characteristics from natural scene and document images.
2. Multi-granularity hierarchical annotations support different application needs.
3. This work represents a substantial data collection effort with 735K images and 4.2M annotations.

**Weaknesses:**

#

1. Limited task scope severely restricts practical value: The dataset provides only bounding box annotations without text transcriptions, which prevents end-to-end OCR applications: Modern OCR datasets all provide both localization and recognition annotations. Text detection alone is merely a preliminary task in the OCR pipeline. While the authors acknowledge this limitation, this is a fundamental requirement for a dataset paper in 2025, not an optional extension.
2. Contradicts the motivation of supporting MLLMs: The authors cite Qwen2.5-VL and Gemma 3 to motivate the work, but these VLMs require more than just bounding boxes. Furthermore, models like Monkey-OCR and Qwen2.5-VL have already achieved strong performance on artistic fonts, raising the question: what unique value does a detection-only dataset provide?
3. The paper presents contradictory evidence on task difficulty. Table 5 shows a puzzling result:

    > YOLOv11 (ICDAR15 → ICDAR15):  F1 = 0.565, mAP₅₀:₉₅ = 0.291
    >

    > YOLOv11 (AnimeText → AnimeText): F1 = 0.851, mAP₅₀:₉₅ = 0.806
    >

    The model performs significantly better on AnimeText. The results show that AnimeText may actually be an easier task than natural scene text detection.

4. There is insufficient analysis of anime text as artistic/stylized fonts. The paper repeatedly mentions "handwritten and stylized artistic fonts" as a key characteristic, but provides no comparison with other datasets containing artistic fonts.

**Questions:**

1. What is the quantitative comparison with Manga109?
2. How does anime text differ from other artistic text domains?

**Details Of Ethics Concerns:**

While authors claim images are "compliant with research-use licenses," anime content involves complex copyright issues.  "Online sources" may include unauthorized fan distributions rather than licensed content.

---

> ### Author Response · Authors · 2025-11-25
>
> ## W1
>
> **Q: Limited task scope (detection only) vs. End-to-end models.**
>
> **Q: Unique value vs. Monkey-OCR/Qwen2.5-VL?**
>
> A: Anime text detection has standalone value and addresses specific bottlenecks:
>
> 1. **Independent Utility:** Detection enables tasks independent of recognition, such as **automated text removal (inpainting)** and data cleaning. Precise **polygon-level annotations** are critical for background restoration in translation workflows, which recognition-based models cannot provide.
> 2. **Technical Bottleneck:** As shown in Sec. 3 and Table 5, general detectors fail on anime's diverse forms and irregular layouts. Robust localization is a strictly necessary prerequisite for effective recognition in this domain.
>
> ## W2
>
> **Q: Contradiction with MLLMs (Qwen, Gemma).**
>
> A: Detection complements rather than duplicates MLLMs:
>
> 1. **Speaker-Text Association:** MLLMs lack spatial grounding for complex layouts. Bounding boxes are vital to map speech bubbles to specific characters, facilitating narrative understanding.
> 2. **Hard Attention Mechanism:** Complex anime backgrounds distract full-image inference. A **"Detect-then-Crop"** strategy acts as hard attention. As shown below, this filters background noise, drastically reducing DeepSeek-OCR’s NED from ​**0.2961 to 0.1030**​.
>
> | Method               | NED ↓ |
> | --- | --- |
> | DeepSeek-OCR     | 0.2961     |
> | DeepSeek-OCR w/ AnimeText crop | 0.1030   |
>
> ## W3
>
> **Q: Contradictory evidence (AnimeText appears "easier").**
>
> A: High performance stems from data scale, not task simplicity:
>
> 1. **Scaling Laws:** Our dataset is >300x larger than ICDAR15. Reducing training data to 35k causes a drastic drop (mAP ​**0.806 **\$\\to\$** 0.712**​), proving results are driven by large-scale supervision.
>
> | **Image Count** | **Precision** | **Recall** | **F1-score** | **mAP@0.5:0.95** |
> | ----------------------- | --------------------- | ------------------ | -------------------- | ------------------------ |
> | 735k                  | 0.878               | 0.825            | 0.851              | 0.806                  |
> | 35k                   | 0.751               | 0.736            | 0.743              | 0.712                  |
>
> 2. **Domain Asymmetry:** AnimeText-trained models generalize to ICDAR15 (F1=0.233), whereas ICDAR15-trained models fail on AnimeText (F1=0.079). This confirms AnimeText contains complex distributions (e.g., severe occlusions) absent in natural scenes.
>
> ## Q2
>
> **Q: Difference from other artistic text?**
>
> A: Anime text poses unique challenges:
>
> 1. **Integration & Occlusion:** Text often blends into background strokes or serves as composition, unlike salient artistic posters.
> 2. **Density & Scale:** Features denser clusters with smaller spatial footprints (e.g., background UI).
> 3. **Low Saliency:** Stylistic blending with illustrations reduces distinctiveness compared to high-contrast advertising text.
>
> ## W4
>
> **Q: Comparison with artistic/stylized fonts.**
>
> A: We compared against MoviePoster (artistic text dataset).
>
> To quantify the difference between anime text and general artistic fonts, we conducted a cross-dataset evaluation using MoviePoster (a representative artistic text dataset).
>
> While MoviePoster models perform well in-domain, they fail on AnimeText (Recall 0.1877). This empirical evidence confirms that AnimeText’s distribution (complex integration with illustrations) is distinct from general artistic text, making existing artistic detectors insufficient.
>
> | Method   | Train Dataset | Test Dataset | Precision | Recall | F1-score | mAP@0.5:0.95 |
> | ---- | --- | --- | --- | --- | --- | --- |
> | Artistic-style text detector | MoviePoster   | MoviePoster  | 0.8889    | 0.8585 | 0.8734   | 0.7021 |
> | Artistic-style text detector | MoviePoster   | AnimeText    | 0.5481    | 0.1877 | 0.2796   | 0.1296  |
> | YOLO v11  | AnimeText  | MoviePoster  | 0.8264    | 0.6218 | 0.7096   | 0.5933   |
>
> ## Q1
>
> **Q: Quantitative comparison with Manga109.**
>
> A: AnimeText is a superset with a significant domain gap.
>
> | Train Dataset | Test Dataset | Precision | Recall | F1-score | mAP@0.5:0.95 |
> | ----- | ----- | ---- | --- | --- | ---- |
> | Manga109      | Manga109     | 0.9453    | 0.9336 | 0.9394   | 0.8215 |
> | Manga109      | AnimeText    | 0.8646    | 0.3985 | 0.5455   | 0.3742  |
> | AnimeText     | Manga109     | 0.8441    | 0.8966 | 0.8696   | 0.6912  |
> | Manga109    | CTW1500      | 0   | 0  | 0   | 0  |
> | Manga109    | ICDAR15      | 0  | 0  | 0   | 0      |
>
>
> Manga109-trained models show **high Precision** but **low Recall (0.40)** on AnimeText.
>
> 1. **Superset:** The low recall indicates \~60% of AnimeText instances (fan-edits, complex scenes) are beyond Manga109's scope.
> 2. **Visual Gap:** Unlike monochromatic static manga, AnimeText includes ​**RGB color**​, ​**dynamic lighting**​, and ​**motion blur**​.
>
> These unique challenges highlight why existing manga-based methods are insufficient and underscore the necessity of our proposed benchmark.

---

### Official Review · Reviewer_mveS · 2025-10-31

**Soundness:** 3
**Presentation:** 3
**Contribution:** 2
**Rating:** 4
**Confidence:** 4

**Summary:**

This paper focuses on addressing the gap in text detection datasets for anime scenes. Current text detection datasets are mainly designed for natural or document scenes. To fill this gap, the authors propose AnimeText, a large-scale dataset consisting of 735K images and 4.2M annotated text blocks. This dataset is equipped with hierarchical annotations and hard negative samples, both specifically tailored to meet the needs of anime scenarios. For evaluating the dataset, this paer conducts cross-dataset benchmarking using state-of-the-art text detection methods. Experimental results indicate that models trained on AnimeText perform better than those trained on existing datasets when handling text detection tasks in anime scenes, which confirms the dataset's effectiveness in addressing the aforementioned gap.

**Strengths:**

1) Using current datasets to train text detectors can not address the anime text detection problem well, so constructing a specialized anime text detection dataset is helpful.
2) The scalable annotation workflow is time-consuming and the evaluation is useful for subsequent anime text detection research.
3) Cross-dataset experiments and ablation studies are implemented to demonstrate the effectiveness of the proposed dataset.

**Weaknesses:**

1) A baseline anime text detection method is absent in the paper. Only using previous detection model to show the effectiveness of the dataset is not enough. What is the problem that scaling data can not solve?
2) From Tab.5, the F1-Score of anime text detection is superior to that of IC15. Is it overfitting? Or just the training samples work?

**Questions:**

See Weaknesses.

---

> ### Author Response · Authors · 2025-11-25
>
> ## W1
>
> **Q: A baseline anime text detection method is absent ​**​**in the**​**​ paper. Only ​**​**using**​**​ previous detection model to show the effectiveness of the dataset is not enough.**
>
> A: We clarify that as the first anime text detection dataset, adapting SOTA text detectors serves as a rigorous baseline. While existing manga datasets focus on static, black-and-white content, our dataset acts as a **superset** with a significant ​**domain gap**​:
>
> 1. **Content Diversity:** Beyond comics, our dataset integrates AI-generated ​**anime images**​, screenshots, **edited anime, ​**posters, and derivative fan works.
> 2. **Visual Gap:** Unlike **monochromatic** manga scans, anime images usually feature complex ​**RGB color spaces**​, dynamic ​**lighting/shading**​, and video-specific anomalies like **motion blur** and compression artifacts.
> 3. **Text Layout:** Anime text can lacks speech bubbles and features extreme artistic stylization (e.g., heavy strokes, glow effects).
>
> These unique challenges highlight why existing manga-based methods are insufficient and underscore the necessity of our proposed benchmark.
>
> We further provide the comparison of AnimeText with manga109 (SOTA manga dataset),  shown in the table below:
>
> | Train Dataset | Test Dataset | Precision | Recall | F1-score | mAP@0.5:0.95 |
> | --------------- | -------------- | ----------- | -------- | ---------- | -------------- |
> | Manga109      | Manga109     | 0.9453    | 0.9336 | 0.9394   | 0.8215       |
> | Manga109      | AnimeText    | 0.8646    | 0.3985 | 0.5455   | 0.3742       |
> | AnimeText     | Manga109     | 0.8441    | 0.8966 | 0.8696   | 0.6912       |
> | Manga109      | CTW1500      | 0         | 0      | 0        | 0            |
> | Manga109      | ICDAR15      | 0         | 0      | 0        | 0            |
>
> The model trained on Manga109 achieves **high Precision (0.86)** but **low Recall (0.40)** on AnimeText. These results perfectly illustrate our point above.
>
> **Q:What is the problem that scaling data can not solve?**
>
> A: As discussed in Sec. 4.2 and Sec. 6.3. While data scaling significantly improves general detection capabilities, it yields diminishing returns on ​**hard samples with high visual ambiguity**​. Scaling data volume alone helps the model memorize patterns but often fails to resolve these fine-grained semantic confusions without specific architectural designs or hard-negative mining. Therefore, we specifically annotated these hard samples to establish a benchmark, facilitating future research to develop methods that go beyond simple data scaling.
>
> ## W2
>
> **Q: From Tab.5, the F1-Score of anime text detection is superior to that of IC15. Is it overfitting? Or just the training samples work?**
>
> A: The high F1-Score is attributed to the ​**scale and diversity of the data**​, not overfitting. We clarify this in three aspects:
>
> 1. **Strictly Disjoint Splits:** Our training and testing sets are constructed from different sources with no overlap, preventing data leakage.
> 2. **Efficacy of Data Scaling:** Consistent with neural scaling laws, the performance gain stems from the massive increase in training data volume. As shown in the table below, reducing the training set from 735k to 35k results in a significant performance drop, confirming that the high score is data-driven.
>
> | Image Count | Precision | Recall | F1-score | mAP@0.5:0.95 |
> | ------------- | ----------- | -------- | ---------- | -------------- |
> | 735k        | 0.878     | 0.825  | 0.851    | 0.806        |
> | 35k         | 0.751     | 0.736  | 0.743    | 0.712        |
>
> 3. **Generalization:** Overfitting typically leads to poor performance on unseen domains. However, as shown in ​**Tab. 5**​, the model trained on our AnimeText dataset demonstrates **stronger generalization capabilities** on other out-of-domain datasets compared to models trained on ICDAR15 and CTW-1500. This further validates that the model has learned robust features rather than memorizing the training set.

---

### Official Review · Reviewer_wFPr · 2025-11-02

**Soundness:** 3
**Presentation:** 2
**Contribution:** 2
**Rating:** 4
**Confidence:** 5

**Summary:**

This paper introduces the AnimeText dataset, the first large-scale dataset (735K images, 4.2M text blocks) for anime scene text detection, addressing the lack of benchmarks in stylized 2D visual contexts.

**Strengths:**

The AnimeText dataset is the first large-scale benchmark for anime scene text detection (735K images, 4.2M text blocks), filling a key research gap where existing datasets focus on natural or document images. It is substantially larger than prior benchmarks (26× TextOCR, 72× ICDAR19-ArT) and supports multiple languages, including Japanese, English, Chinese, Korean, and Russian.  It features hierarchical polygon-level annotations suitable for fine-grained detection and OCR, and a three-stage construction pipeline combining manual labeling, CLIP-based hard-negative filtering (Acc/F1 ≈ 98%), and multi-granularity tagging. Benchmarks across YOLOv11, DBNet, LRANet, and Bridging Text Spotting demonstrate major performance gains (F1 ↑ from 0.008 to 0.855 for LRANet).

**Weaknesses:**

1. While B0/B1 annotation levels are defined, the paper does not specify which level is used for cross-dataset evaluation in Table 5. Since AnimeText employs a coarser, block-level labeling scheme, direct comparison with word-level datasets such as ICDAR15 is not strictly fair or interpretable.

2. No controlled experiments are provided to isolate the effects of annotation granularity on model performance (e.g., by splitting B0 regions into word-level annotations or merging ICDAR15 into block-level). Consequently, it is difficult to disentangle performance differences caused by domain gap versus annotation gap.

3. The paper shows limited comparison with structurally similar datasets.  Although CTW-1500 appears in the dataset statistics and uses polygonal, line-level annotations (one box per text line, including whitespace), it is omitted from experimental comparisons. Given its structural similarity to AnimeText’s B0 format, CTW-1500 would serve as a more aligned and informative baseline than ICDAR15.

4. The reported precision gains from hard-negative filtering are not accompanied by statistics on mistakenly removed true text instances (false negatives). This omission obscures the impact on recall and overall F1 performance.
5. While the paper notes that AnimeText exhibits higher text density and peripheral text positioning, it stops short of quantifying how these factors affect detection performance (e.g., through density-stratified F1 curves or ablation analysis).

6. All data originate from online anime sources under research-use licenses. The absence of real-world or user-generated images (e.g., screenshots, edited anime, or derivative fan content) limits the dataset’s robustness and generalization potential for practical applications.

7. AnimeText focuses solely on text localization without providing transcription-level annotations. This restricts its applicability for full OCR pipelines, subtitle translation, or vision-language understanding tasks (e.g., VQA).

8. Several contemporary foundation-level or transformer-based text detection models are omitted from comparison. Including such systems would provide a stronger contextual benchmark and clarify where AnimeText stands relative to current state-of-the-art detectors.

**Questions:**

1. Which annotation level (B0 or B1) is used for the cross-dataset evaluation in Table 5? If B1 annotations are used, how is fairness ensured when comparing to word-level datasets such as ICDAR15, given the coarser granularity of AnimeText? Conversely, if B0 annotations are used, what specific rules govern multi-line grouping, intra-line whitespace handling, and segmentation consistency across varying text layouts?

2. While the paper reports a +26.9 % increase in precision and improvements in Acc/F1 at the classifier level, it does not disclose the false-negative rate—i.e., the proportion of true text instances mistakenly filtered out. Could the authors provide quantitative results on the percentage of true text erroneously removed and its subsequent impact on recall and overall F1 ? Such information would help assess whether the precision gains come at the expense of text coverage.

3. Can the authors experimentally isolate the impact of annotation granularity? For example:  a. Split AnimeText B0 annotations into word-level units and re-evaluate performance. b. Merge ICDAR15 word-level boxes into block-level regions and report mAP/F1 for both settings.
This would help disentangle differences attributable to annotation granularity from those caused by domain variation.

4. Why was CTW-1500 excluded from comparative evaluation? As CTW-1500 provides line-level polygon annotations, structurally similar to AnimeText’s B0 format, it would serve as a more directly comparable dataset than ICDAR15 and help validate cross-annotation consistency.

5. AnimeText consists exclusively of static frames and omits temporal continuity information relevant to anime videos—such as motion blur, scene transitions, or persistent subtitle overlays.  This limitation significantly reduces applicability to real-world anime text spotting and temporal OCR pipelines.

**Details Of Ethics Concerns:**

No.

---

> ### Author Response · Authors · 2025-11-25
>
> ## W1, W2, Q1, Q3
>
> ***Q: the paper does not specify which level is used for cross-dataset evaluation in Table 5. ​***
>
> A: We clarify that **\$B^0\$ annotation levels** were used for the evaluation in Table 5.
>
> ***Q: AnimeText uses block-level labels and ICDAR15 uses word-level labels.  It is difficult to disentangle performance differences caused by domain gap versus annotation gap.***
>
> A: Thank you for the comment. We clarify that to ensure a fair cross-dataset evaluation despite the format difference (block-level in AnimeText vs. word-level in ICDAR15), we merged the word-level labels of ICDAR15 into block-level labels using the method described in ​**Sec 4.3**​. This preprocessing eliminates the annotation gap, ensuring that the performance differences reported in Table 5 stem primarily from the domain gap between natural and anime scenes. We emphasized this trick in the revised manuscript.
>
> ## W3, Q4
>
> ***Q: The paper shows limited comparison with structurally similar datasets. ... Given its structural similarity to AnimeText’s B0 format, CTW-1500 would serve as a more aligned and informative baseline than ICDAR15.***
>
> A: We agree that CTW-1500 is a valuable baseline due to its structural similarity. We have performed additional experiments using YOLO v11 to compare CTW-1500, ICDAR15, and AnimeText (Table 5 in the revised main paper).
>
> | Method   | Train Dataset | Test Dataset | Precision | Recall | F1-score | mAP@0.5:0.95 |
> | ---------- | --------------- | -------------- | ----------- | -------- | ---------- | -------------- |
> | YOLO v11 | CTW1500       | CTW1500      | 0.8072    | 0.7901 | 0.7986   | 0.5503       |
> | YOLO v11 | CTW1500       | AnimeText    | 0.2335    | 0.2673 | 0.2493   | 0.0698       |
> | YOLO v11 | AnimeText     | CTW1500      | 0.4224    | 0.3238 | 0.3666   | 0.1754       |
> | YOLO v11 | CTW1500       | ICDAR15      | 0.4579    | 0.3722 | 0.4106   | 0.1661       |
> | YOLO v11 | ICDAR15       | CTW1500      | 0.5120    | 0.4171 | 0.4597   | 0.2394       |
>
> These results support three key conclusions:
>
> 1. **Consistency with Table 5:** A significant domain gap exists between natural and anime scenes. The model trained on CTW-1500 performs poorly on AnimeText (F1 0.2493).
> 2. **Domain Gap Magnitude:** While a domain gap exists between natural datasets (CTW-1500 \\(\to\\) ICDAR15, F1 0.4106), it is significantly smaller than the gap between natural and anime datasets (CTW-1500 \\(\to\\) AnimeText, F1 0.2493).
> 3. **Benefit of Large Scale:** The model trained on AnimeText shows reasonable generalization to CTW-1500 (F1 0.3666), suggesting the diversity and scale of AnimeText benefit feature learning.
>
> ## W4, Q2
>
> ***Q: The reported precision gains from hard-negative filtering are not accompanied by statistics on mistakenly removed true text instances (false negatives). This omission obscures the impact on recall and overall F1 performance.***
>
> A: Thank you for the comment. We would like to clarify that ​**Table 6 already reports both precision and recall**​, and both metrics consistently improve after applying ​**hard-negative sample filtering**​, suggesting that the filtering does not introduce much false negatives. Furthermore, the filtering is **only used during training** to improve feature discrimination, while the ​**test set evaluation remains unaffected**​.
>
> ## W5
>
> ***Q: While the paper notes that AnimeText exhibits higher text density and peripheral text positioning, it stops short of quantifying how these factors affect detection performance (e.g., through density-stratified F1 curves or ablation analysis).***
>
> A: Thank you for the suggestion. We have conducted a density-stratified analysis, measuring F1-scores across images with varying text densities (Figure I in the revised supplement material). The resulting curves show that while performance slightly decreases for extremely high-density images, the detection performance remains relatively stable across most density levels. This demonstrates the robustness of the dataset and baseline models against varying text distributions.

---

> ### Author Response · Authors · 2025-12-04
>
> ## W6, Q5
>
> ***Q: All data originate from online anime sources under research-use licenses. The absence of real-world or user-generated images (e.g., screenshots, edited anime, or derivative fan content) limits the dataset’s robustness and generalization potential for practical applications.***
>
> A: We clarify that our data collection sources are not limited to raw anime frames. The dataset explicitly includes a wide variety of ​**real-world and user-generated content**​, including screenshots, edited anime images, comics, posters, derivative fan works, and AI-generated images. This diversity is a key advantage of AnimeText over existing manga or comic datasets, which often rely on clean, scanned pages. We have included some examples in revised supplement material.
>
> We empirically prove our practical values by comparing our AnimeText with manga109 (absence of real-world or user-generated images).
>
> | Method   | Train Dataset | Test Dataset | Precision | Recall | F1-score | mAP@0.5:0.95 |
> | ---------- | --------------- | -------------- | ----------- | -------- | ---------- | -------------- |
> | YOLO v11 | Manga109      | Manga109     | 0.9453    | 0.9336 | 0.9394   | 0.8215       |
> | YOLO v11 | Manga109      | AnimeText    | 0.8646    | 0.3985 | 0.5455   | 0.3742       |
> | YOLO v11 | AnimeText     | Manga109     | 0.8441    | 0.8966 | 0.8696   | 0.6912       |
> | YOLO v11 | Manga109      | CTW1500      | 0         | 0      | 0        | 0            |
> | YOLO v11 | Manga109      | ICDAR15      | 0         | 0      | 0        | 0            |
>
> As shown in table above, a model trained on Manga109 achieves **high Precision (0.86)** but **low Recall (0.40)** on AnimeText. This leads to two conclusions:
>
> 1. **AnimeText is a Superset:** The high precision confirms our dataset covers standard clean data, while the low recall proves it extends significantly beyond it. The \~60% missed instances correspond to the complex, real-world and user-generated images (e.g., fan-edits, distortions) unique to AnimeText.
> 2. **Robustness:** The manga109 trained model fails to generalize on natural scenes (CTW1500/ICDAR15), confirming that AnimeText's diverse distribution is essential for robust, practical generalization.
>
> ## W7
>
> ***Q:AnimeText focuses solely on text localization without providing transcription-level annotations. This restricts its applicability for full OCR pipelines, subtitle translation, or vision-language understanding tasks (e.g., VQA).***
>
> A: As stated in the title and abstract, the primary scope of this work is ​**Anime Scene Text Detection**​. While we acknowledge the importance of OCR and VQA, providing high-quality transcription for anime text (often stylized or invented languages) requires a distinct annotation pipeline.
>
> But we respectfully argue that text detection in the anime domain has significant standalone value for two reasons:
>
> 1. **Independent Utility: ​**Text detection in anime scenes has many practical applications ​**independent of vision-language understanding tasks**​, such as automated text removal, data cleaning for generative models, and speaker-text association. Our dataset is primarily designed to serve these scenarios rather than text understanding tasks.**​ ​**For example, text detection is the critical first step for **automated text removal (inpainting)** in translation workflows. Our dataset provides fine-grained ​**polygon-level annotations**​, which are essential for precise text erasure and background restoration. This application relies entirely on accurate localization, independent of recognition tasks.
> 2. **Technical Bottleneck:** Unlike natural scenes, anime scenes have different challenges as stated in sec.3 (diverse forms,  irregular styles and layouts, confusing symbols and patterns). Existing general-purpose detectors fail heavily on these cases (as shown in Table 5). Robust detection is a prerequisite that must be solved before effective recognition is possible.
>
> We believe AnimeText serves as a solid foundation for the community to build upon for future transcription-level tasks, but extending it to OCR/VQA is outside the scope of this specific contribution.
>
> ## W8
>
> ***Q: Several contemporary foundation-level or transformer-based text detection models are omitted from comparison. Including such systems would provide a stronger contextual benchmark and clarify where AnimeText stands relative to current state-of-the-art detectors.***
>
> A: We would like to clarify that ​**Table 5 already includes "Bridging Text Spotting"**​, which is a representative state-of-the-art, Transformer-based text detection model. Its inclusion provides a benchmark relative to modern architectures.

---

### Meta-Review · Area_Chair_A6JU · 2026-01-07

**Summary:**

This paper introduces AnimeText, a large-scale dataset for anime scene text detection. The reviewer's main concerns can be summarized as follows:

- Reviewer wFPr:
  - Lacks rigor in cross-dataset evaluation due to mismatched annotation granularities, such as with ICDAR15.
  - No controlled experiments are provided to disentangle annotation effects from domain differences.
  - Limited comparisons with structurally similar datasets such as CTW-1500.
  - Incomplete analysis of hard-negative filtering (as its impact on recall is not reported).
  - Dataset-specific characteristics such as high text density and peripheral text placement are not quantitatively analyzed.
  - The dataset scope is limited by reliance on a single data source domain and is further limited to text localization without transcription-level annotations.
  - Limited experiments that omit several modern transformer-based detectors for comparison.

- Reviewer mveS:
   - Lacking a baseline anime text detection method + demonstrating effectiveness using only existing detection models is insufficient.
   - Concerns remain regarding data scaling.

- Reviewer wHE5:
   - Limited (bounding boxes only) task scope that weakens OCR and MLLM relevance.
   - Contradicting the motivation of supporting MLLMs that are provided with annotations beyond bounding boxes.
   - Presence of contradictory evidence on task difficulty, where AnimeText may actually be an easier task.
   - Insufficient analysis of anime text as artistic/stylized fonts (missing comparison with other datasets containing artistic fonts).
   - Missing quantitative comparison with Manga109.
   - Domain difference between anime text vs. artistic text domains.

- Reviewer 7oZU
  - Incomplete discussion of existing datasets (AnimeText is not clearly positioned relative to prior anime or manga text datasets).
  - Weak justification of domain uniqueness (the experiments show a domain gap but do not establish anime as a uniquely distinct domain).
  - Insufficient evidence for MLLM relevance (Claims of MLLM relevance are unsupported by the detection-only evaluation).

**Reviewer Concerns:**

The authors submitted an AC message including concerns regarding the likelihood of LLM reviews by Reviewer 7oZU. The AC has reviewed the paper individually and carefully considered all content for the final decision, which is based on an overall assessment of the paper and the full set of reviews, rather than on any single review in isolation.

The shared critical concerns can be classified into about five classes: (1) limited task scope and overstated multimodal LLMs (MLLMs) relevance; (2) insufficient positioning against existing datasets; (3) weak justification of domain uniqueness;  (4) insufficient analysis of dataset-specific characteristics; (5) incomplete experimental evaluation with some concerns about evaluation reliability.

This AC largely agrees with the reviewers' concerns and would like to provide constructive feedback that the authors are encouraged to address in future submissions. A dataset paper should include a more comprehensive literature survey and detailed comparisons with closely related datasets, rather than focusing primarily on datasets that are relatively irrelevant, such as standard scene-text datasets, as in Table 2. Although Manga109 was cited as related work, it was not compared in sufficient detail, as noted by the reviewers as well. As a result, it is still unclear what additional value the proposed dataset provides beyond existing anime-style text datasets.

Second, the scope of the dataset is inherently limited. While the paper stresses supporting MLLMs as a key motivation, only bounding box annotations constrain the dataset to text detection without recognition, substantially limiting its practical value for MLLM use cases. Although the authors suggest applicability to MLLMs, the paper does not provide concrete experimental evidence demonstrating value beyond bounding-box supervision. Moreover, the claimed domain uniqueness is not sufficiently clarified, as it remains unclear whether one could observe differences that reflect anime-specific properties, more general artistic or stylized text characteristics, or typical discrepancies between anime scenes and standard scene-text datasets. Consequently, whether due to the presentation or the scope definition, the dataset's overall positioning is not convincingly established.

Third, the evaluation protocol would benefit from more rigorous and controlled experiments. The evaluation should provide strong evidence that training on the proposed dataset enables models to address challenges that are difficult to capture with existing datasets. In this regard, the experiments should demonstrate compatibility or complementarity with existing datasets, rather than primarily highlighting a generic domain gap as in Table 5, which does not convincingly justify the dataset's uniqueness. For example, analyzing accuracy trends across scene images when trained on the proposed dataset versus other datasets could offer more informative statistical evidence. The authors are encouraged to further consider evaluation strategies that clearly justify why the community should adopt this dataset. The use of older text-detection models is a relatively minor concern.

**Reviewer Scores:**

All reviewer scores were below the acceptance threshold at the initial stage. Reviewer 7oZU participated in the discussion and indicated that their rating should not be changed, whereas the remaining reviewers did not participate. This AC believes that, even if further discussion were to continue, the paper's score would be unlikely to change, as the significant weaknesses identified above are inherent and cannot be adequately addressed within the constraints of the rebuttal period.

---

### Decision · Program_Chairs · 2026-01-26

Reject